# Photo-Antibacterial Activity of Two-Dimensional (2D)-Based Hybrid Materials: Effective Treatment Strategy for Controlling Bacterial Infection

**DOI:** 10.3390/antibiotics12020398

**Published:** 2023-02-16

**Authors:** Neetu Talreja, Divya Chauhan, Mohammad Ashfaq

**Affiliations:** 1Department of Science, Faculty of Science and Technology, Alliance University, Anekal, Bengaluru 562106, India; 2Department of Drinking Water and Sanitation, Ministry of Jal Shakti, New Delhi 110003, India; 3Department of Biotechnology, University Centre for Research & Development, Chandigarh University, Gharaun, Mohali 140413, India

**Keywords:** photo-antibacterial agent, 2D-NMs, photocatalysis, environmental remediation, infectious disease, artificial intelligence, carbon

## Abstract

Bacterial contamination in water bodies is a severe scourge that affects human health and causes mortality and morbidity. Researchers continue to develop next-generation materials for controlling bacterial infections from water. Photo-antibacterial activity continues to gain the interest of researchers due to its adequate, rapid, and antibiotic-free process. Photo-antibacterial materials do not have any side effects and have a minimal chance of developing bacterial resistance due to their rapid efficacy. Photocatalytic two-dimensional nanomaterials (2D-NMs) have great potential for the control of bacterial infection due to their exceptional properties, such as high surface area, tunable band gap, specific structure, and tunable surface functional groups. Moreover, the optical and electric properties of 2D-NMs might be tuned by creating heterojunctions or by the doping of metals/carbon/polymers, subsequently enhancing their photo-antibacterial ability. This review article focuses on the synthesis of 2D-NM-based hybrid materials, the effect of dopants in 2D-NMs, and their photo-antibacterial application. We also discuss how we could improve photo-antibacterials by using different strategies and the role of artificial intelligence (AI) in the photocatalyst and in the degradation of pollutants. Finally, we discuss was of improving the photo-antibacterial activity of 2D-NMs, the toxicity mechanism, and their challenges.

## 1. Introduction

Bacterial contamination causes infectious diseases and is the third-greatest cause of mortality and morbidity. Bacterial contamination in water bodies is a severe scourge that affects human health globally. Numerous bacterial strains have a critical role in bacterial infectious diseases, but our main focus is bacterial infections caused by contaminated water. Contaminated water is a severe global threat [1,2,3,4,5]. High levels of contamination, either chemical or biological, in water cause severe health issues. Biological contamination, mainly bacteria and viruses, should be eradicated from water bodies to provide safe and healthy water to society. Biological contamination is responsible for many waterborne and infectious diseases. These bacteria have developed resistance against antibiotics because of excessive use of antibiotic drugs. Bacterial resistance affects bacterial strains’ tolerance ability, leading to severe health issues in human beings [6,7,8,9]. For this reason, newer materials that effectively remove biological contamination from water bodies and at the site of infection need to be developed. In this sense, nanomaterials might kill/inhibit bacterial infection without causing increased bacterial resistance.

Nanomaterials have gained increasing interest from researchers for developing newer antibiotic materials. We have many reasons to believe that nanomaterials play an essential role in controlling bacterial infections, as nano-sized materials easily penetrate bacterial cell walls. For instance, there are many nanomaterials using Cu, Zn, Ag and Au, as well as CNTs, CNFs, and two-dimensional nanomaterials (2D-NMs), that have been used as antibiotic materials that effectively kill/inhibit bacterial strains, especially those with multi-drug resistance (MDR) and extensive drug resistance (XDR) [10,11,12,13,14,15,16,17,18,19,20,21,22]. Another way to reduce or minimize MDR and XDR is via photo-antibacterial activity, which offers newer possibilities for controlling bacterial infections from water using photocatalytic materials, cellular oxygen, and visible light irradiation [23,24,25,26]. Moreover, it is important to mention that these materials have insignificant toxicity against different cell lines, whereas some materials exhibit some toxicity at a certain level [27,28,29,30]. Therefore, photo-antibacterial activity is an exceptional candidate for controlling bacterial infection at the site of infection and from water bodies.

Numerous semiconductor materials such as TiO_2_, ZnO, CdS, C_3_N4, BiOX, and 2D-NMs have been effectively used in the photocatalysis process to effectively degrade various pollutants, including bacteria. Among them, 2D-NM-based semiconductor materials, such as layered double hydroxides (LDHs), graphitic carbon nitride (g-C_3_N_4_), black phosphorus (BP), graphene, graphene oxide (GO), and transitional metal chalcogenides (TMDs) (WS_2_, MoS_2_, MXene, TeS_2_), effectively degrade various pollutants and kill/inhibit bacteria. Usually, 2D-NMs are a class of nanomaterials with a one to two atomic layer thickness, giving them exceptional optical, electrical, and mechanical properties. The 2D-NM-based semiconductor materials are more compelling than their bulk materials for use as photo-antibacterial agents due to their irreplaceable characteristics such as high surface area, high electrical and mechanical stability, functionalization ability, tunable band gap, and high biocompatibility [31,32,33,34,35,36,37,38,39,40]. Usually, these 2D-NMs generate electron–hole pairs and charge separation upon exposure to solar irradiation. The produced photocarriers mainly generate reactive oxygen species (ROS), hydroxyl radicals, and superoxide radicals as reacting species for the degradation of pollutants and killing/inhibiting of bacteria during photocatalytic activity. Moreover, incorporating metals, carbon, polymers, and surface functionalization within 2D-MNs (or 2D-NM-based hybrid materials) improves their band gap value, biocompatibility, and photon absorption, producing high photo-antibacterial activity [37,41,42]. In this sense, 2D-NM-based hybrid materials have numerous advantages in terms of high photo-antibacterial activity, high biocompatibility, and being environmentally friendly.

Furthermore, 2D-NM-based hybrid materials provide a newer platform with advancement in the materials sciences, with which we can easily tune structural, optical, and electronic properties. Usually, photo-antibacterial agents depend on the photocatalyst material’s characteristics, such as crystallinity, size, shape, surface area, and porous texture. Dosage, pH, pollutant concentration, and light intensity affect photocatalytic activity as well. Undoubtedly, synthesizing 2D-NM-based hybrid materials using different components with different properties might be beneficial for developing photo-antibacterial agents with desired properties [43,44]. However, enumerating the usefulness of photocatalysts in controlling bacterial infection is one of the most significant challenges of today. We can resolve such associated issues with the help of artificial intelligence (AI), which can effectively design experiments and optimize the photocatalyst materials, thereby minimizing the cost of experiments [45,46,47]. This review article focuses on the photo-antibacterial activity of 2D-NMs and their hybrid materials. We discuss the effects of dopants such as metals, polymers, and 2D-NMs within the 2D-NMs on photo-antibacterial activity. Additionally, biocompatibility, mode of action, and the role of AI in photo-antibacterial activity are also discussed. Finally, challenges and future aspects of 2D-NMs and 2D-NM-based hybrid materials are also discussed, which provide newer insight into photo-antibacterial activity (Figure 1).

## 2. Photo-Antibacterial Activity

Bacterial infectious diseases are one of the severe threats that affect human beings. There are several strategies, such as the use of chemically and naturally synthesized antibiotics, to inhibit/kill bacterial strains. Metal and metal-oxide-based nanomaterials are proven antibacterial agents that effectively kill/inhibit all bacterial strains (including MDR and XDR strains). Usually, the development of bacterial resistance ensues in one of several ways, including (1) alteration in interface/interaction site between bacteria and antibiotics, (2) leakage of antibiotics to reduce concentration in bacteria, and (3) alteration of antibiotics or enzymatic degradation. Several treatment strategies, such as microwaveocaloric, sonodynamic, and photo-responsive, have been efficiently used to treat bacterial infections.

Furthermore, these treatment strategies rapidly kill/inhibit bacterial strains compared to conventional methods. The photo-responsive or photo-antibacterial strategy is continuously gaining interest due to the easy and cost-effective nature of the technology. Usually, photo-antibacterial activity is produced by photocatalytic materials that activate by absorbing photons to generate ROS, damaging bacterial cellular membranes and thereby killing/inhibiting bacteria. The ROS enter within the bacterial cells that destroy lipids and proteins, thereby causing cellular disruption of the bacterial cells and stopping other physiological activity. Photo-antibacterial activity has several advantages, such as relatively faster response and minimized/avoided bacterial resistance compared with conventional antibiotic treatment [48,49,50,51]. Usually, photo-antibacterial activity involves the use of semiconductor materials to kill/inhibit bacteria, whereas antibiotic molecules might lead to bacterial resistance by encouraging the survival of resistant bacterial strains. The photo-antibacterial agents kill bacteria by damaging the cellular membranes/other structures, thereby impeding the development of bacterial resistance. Moreover, photo-antibacterial agents often have a broader spectrum of antibacterial activity compared with antibiotic molecules, further impeding bacterial resistance. It is important to mention here that the development of bacterial resistance is a complex process and occurs with prolonged or excessive use of any antibacterial agents [48,52,53]. Figure 2 shows a graphical illustration of the photo-antibacterial activity of nanomaterials. Furthermore, photo-antibacterial agents have high antibacterial activity against gram-positive bacteria compared with gram-negative bacteria due to their cell wall structures. Gram-negative bacteria have an additional outer membrane that is difficult to penetrate, whereas gram-positive bacteria are composed of a loose peptidoglycan layer that is easily penetrated by small molecules or is more sensitive to ROS. Photo-antibacterial agents are a broad-spectrum therapeutic strategy because photo-excited ROS affect many metabolic pathways of bacterial strains that depend on their cellular structure. Moreover, immediate ROS generation might easily kill bacteria under solar irradiation, thereby not causing bacterial resistance. 

## 3. Role of Oxygen in Photo-Antibacterial Activity

Bacterial infectious disease is one of the most severe global threats due to bacterial resistance. Indeed, researchers continue to focus on newer antibiotic materials, especially 2D-NMs, that efficiently kill/inhibit bacterial strains without developing bacterial resistance due to their exceptional characteristics. Photo-induced activity is one ancient technique used for environmental pollutant degradation. Researchers have observed that ambient oxygen is necessary to initiate the reaction for the degradation of pollutants and inactivation of bacterial strains. Solar irradiation exposure effectively kills numerous bacterial strains because of the photo-stimulation of endogenous porphyrins. Usually, two types of photo-induced reactions kill the bacterial strains: (1) Type-1 pathways, which comprise electron transfer reactions, lead to the generation of ROS (hydroxyl radicals, superoxide, and hydrogen peroxide). Excess ROS leads to the cellular disruption of bacterial strains, thereby causing cell death of the bacteria. (2) Type-2 pathways comprise the energy transfer from photosensitizers that can generate singlet oxygen. It is essential to mention that other pathways (Type-3), such as Psoralens photo-activation by UV light, generate DNA mono adducts and inter-strand cross-links, subsequently killing bacterial strains. These pathways more effectively kill bacterial strains in the absence of oxygen [54,55,56]. In general, we can say that photo-antibacterial activity is mainly oxygen-dependent, but might be possible in the absence of oxygen. We have substantial evidence suggesting oxygen’s role in photo-antibacterial activity. For instance, Feuerstein et al. use *Porphyromonas gingivalis* and *Fusobacterium nucleatum* in the presence of oxygen under visible light irradiation. Their data suggested that photo-toxicity depends on oxygen, and hydroxyl radicals play an essential role in photo-antibacterial activity [57]. Maclean et al. used *S. aureus* bacterial strains in the presence and absence of oxygen under visible light. Their data indicated a slight decrement in bacterial growth under visible light in the presence of oxygen, which was attributed to oxygen being required for the photo-activation process [58]. Dai et al. used methicillin-resistant *S. aureus* (MRSA) in the presence of oxygen under blue light irradiation. Their data suggested that the reduction of bacterial strains under blue light irradiation and photo-activation is oxygen-dependent [59]. These studies indicate that photo-activation is oxygen-dependent and that ROS plays a crucial role in the inactivation of bacterial cells. Moreover, solar irradiation shows some photo-antibacterial activity in some bacterial strains at a certain level. By using photosensitizers or 2D-NMs, we can easily improve ROS production, leading to bacterial cellular disruption. Photocatalysts based on 2D-NMs mainly operate via the excitation of electron–hole pairs, separation of charge, redox reaction on catalyst surfaces, and generation of ROS, subsequently causing bacterial cell death. It is important to mention here that the the active species varies with different photocatalyst materials.

## 4. Photo-Antibacterial Activity of 2D-NMs

Antibacterial 2D-NMs are among the most promising and frequently used agents due to their remarkable characteristics, high surface area, excellent biocompatibility, cost-effectiveness, and remarkable photocatalytic activity, which make them exceptional materials for photo-antibacterial activity. Moreover, their photo-antibacterial activity with relatively less resistance, rapid sterilization, and non-invasive antibacterial activity makes them outstanding candidates for next-generation photo-antibacterial agents. Additionally, the photo-antibacterial activity might be tuned by different synthesis strategies by changing their morphology and band gap value. Numerous 2D-NMs have been efficiently used for photo-antibacterial activity. For instance, Huang et al. synthesized graphitic carbon nitride (g-C_3_N_4_) using a simple heating process and tested it against an *E. coli* bacterial strain. Their data indicated that the g-C_3_N_4_ produces photo-antibacterial activity with 100% inhibition at 4h of exposure. Moreover, g-C_3_N_4_ is reusable for up to 3 consecutive cycles [60]. Mao et al. synthesized BP using a simple exfoliation process and tested it against both *E. coli* and *S. aureus* under solar irradiation. Their data indicated that the prepared BP effectively kills/inhibit both *E. coli* and *S. aureus* within 10 min of solar irradiation [61]. Markovic et al. synthesized different graphene-based materials such as GO, graphene quantum dots (GQDs), carbon quantum dots (CQDs), and nitrogen-CQDs (N-CQDs) using the Hummers and thermal decomposition processes. The prepared graphene-based materials killed/inhibited the 19 bacterial strains. However, GO shows relatively poor antibacterial activity compared with other graphene-based materials [26]. Zhao et al. synthesized MoS_2_ using the Li-ion intercalation method and tested it against MDR *E. coli* and methicillin resistance *S. aureus* (MRSA) under solar irradiation. Their data suggested that the single layer MoS_2_ effectively controlled bacterial infection. Interestingly, single-layer MoS_2_ showed the complete inhibition of MDR *E. coli* within 15 min of solar irradiation, whereas complete inhibition of MRSA was observed within 25 min of solar irradiation. Additionally, the generation of ROS and smaller size promoted the antibacterial activity [62]. Zhang et al. synthesized MoS_2_ using ultra-sonication, hydrothermal, and intercalation processes for rapid sterilization. Their data showed that MoS_2_ that was prepared using ultra-sonication, hydrothermal treatment, and intercalation effectively killed/inhibited *E. coli* at rates of 33%, 62%, and 99%, respectively, within 180 min of solar irradiation [63]. Wu et al. synthesized MoS_2_ using a simple annealing process, and modulation of the S compound within the MoS_2_ via changing the annealing temperature significantly affected the photocatalytic activity. The prepared MoS_2_ sample severely damaged the bacterial cell wall by increasing the production of ROS due to decrease in the photo-generated electrons [64]. Table 1 shows the different 2D-NMs and their photo-antibacterial activity. The data mentioned above and shown in Table 1 demonstrate that the 2D-NMs can be effectively used for photo-antibacterial activity due to their band gap values, high surface area, and exceptional electronic characteristics. The high surface area improved the active sites, and the lower band gap value improved the photon absorption, resulting in high photo-antibacterial activity. However, as controlling bacterial infection within a few minutes remains a challenge. Researchers continue to focus on developing newer 2D-NMs or amended dopants with existing materials that may be able to control bacterial infections within a few seconds or minutes of solar irradiation. 

### 4.1. Photo-Antibacterial Activity of Metal-Doped 2D-NMs

As an emerging material, 2D-NMs have expanded into many research areas, including biomedical applications, due to their exceptional characteristics. These extraordinary properties, including photo-absorption and lower band gap values, make them good candidates for photo-responsive treatments that can be implemented with solar irradiation to generate ROS for controlling bacteria. Researchers still focus on improving photo-absorption ability and narrowing the band gap values of 2D-NMs, which significantly improves photocatalytic efficiency. In this respect, metal doping within the 2D-NMs might enhance photo-absorption. Usually, the integration of metals within the 2D-NMs enhances active sites, identical pore size distribution, and narrowing of the band gap values, resulting in high photo-responsive ability. Numerous metal-doped 2D-NMs have been synthesized and effectively used for controlling bacterial infection under solar light irradiation. For instance, Ma et al. synthesized Ag–C_3_N_4_-based composite materials using a thermal polymerization process and tested them against *E. coli* under solar irradiation. Their data indicated that the prepared Ag–g-C_3_N_4_-based composite materials had significantly higher photo-antibacterial activity than the pure g–C_3_N_4_. The prepared Ag–g-C_3_N_4_-based composite materials effectively killed/inhibited *E. coli* within 20 min of solar light irradiation. This enhanced photo-antibacterial activity was primarily due to the absorption of photons and rapid transportation of the electron–hole pair [65]. Li et al. synthesized Ag_2_WO_4_–g-C_3_N_4_-based composite materials using a deposition–precipitation process on inactivate bacteria. Their data suggested that the prepared Ag_2_WO_4_–g-C_3_N_4_-based composite materials effectively killed/inhibited bacteria under solar irradiation of 90 min due to the synergetic effect of Ag_2_WO_4_ and g-C_3_N_4_, which improved the separation of electron–hole pairs [66]. Sun et al. synthesized GO–g-C_3_N_4_-based photocatalyst materials and tested them against *E. coli* using a sonication process. Their data indicated that the synthesized GO–g-C_3_N_4_ materials effectively killed the bacteria. Additionally, the GO–g-C_3_N_4_ materials could effectively be used for up to four consecutive cycles at more than 90% inhibition [67]. Kumar et al. synthesized Ag-decorated WS_2_ nanosheets (Ag–WS_2_) using the CVD process and tested them against *E. coli* bacterial strains. Their data suggested that incorporating Ag within the WS_2_ enhanced photo-antibacterial activity, resulting in next-generation materials for fast and smart water-treatment technology [68]. Figure 3 shows a schematic illustration of the photo-antibacterial activity of Ag–WS_2_. Liu et al. synthesized GQDs and ZnO–GQDs using a simple heating process and tested photo-antibacterial activity against both *E. coli* and *S. aureus* bacterial strains. Their data indicated that incorporating ZnO within the GQDs significantly enhanced the photo-antibacterial activity, and complete inhibition/killing was observed at 5 min of exposure [69].

Kiani et al. synthesized rGO-decorated CuO nanowires (rGO–CuO–NW) using oxidation and electrodeposition techniques and tested them against *E. Coli* under solar irradiation. Their results showed that the rGO–CuO–NW wires generated ROS under solar irradiation. Excess ROS could elicit cellular disruption, killing the bacterial strains within 150 min of exposure [70]. Wang et al. synthesized vanadate quantum dots (V-QDs) incorporated with a gC_3_N_4_ (V-QDs–g-C_3_N_4_)-based composite to inactivate *Salmonella* and *S. aureus*. Their results showed that both bacterial strains were effectively inhibited within 15 min of solar irradiation [71]. Liu et al. synthesized Si–rGO-based Ag nanoparticles (Ag–Si–rGO) using simple oxidation and oil–water stratification and controlled a bacterial infection within 5 min of exposure [72]. Another study focused on the synthesis of Ag_2_S-decorated exfoliated WS_2_ using in situ growth, testing the material against *E. coli* and *S. aureus*. Their results showed that the strong adhesion of Ag_2_S onto the surface of WS_2_ efficiently enhanced the antibacterial activity. Figure 4 shows the SEM, TEM, EDS, and crystal structure of Ag_2_S, WS_2_, and Ag_2_S–WS_2_. The SEM and TEM images confirm the adhesion of Ag_2_S onto the surface of WS_2_. The synthesized Ag_2_S–WS_2_ effectively killed/inhibited both *E. coli* and *S. aureus* at 20 min of irradiation; suggesting that Ag_2_S–WS_2_ is a highly effective disinfectant [73]. Wu et al. synthesized a MnO_2_–g-C_3_N_4_-based composite deposited on a Ti implant and tested it for rapid sterilization under solar irradiation. Their results indicated that the MnO_2_–g-C_3_N_4_-based composite coating effectively inhibited/killed bacterial strains within 20 min of solar irradiation with 99.26% and 99.96% inhibition of *E. coli* and *S. aureus* bacteria, respectively. The killing/inhibition of bacteria was mainly due to the generation of ROS during solar irradiation [74]. Sharma et al. synthesized CeO_2_–GO using a simple co-precipitation process and tested it against different bacterial strains. Their data suggested that the CeO_2_–GO effectively inhibited bacterial strains (*E. coli*, *S. aureus*, *S. typhi*, and *P. aeruginosa*.) under solar irradiation. Additionally, the killing/inhibition of the bacterial strains was mainly due to membrane leakage and ROS [75]. Caires et al. synthesized Ag nanoparticles using green synthesis and rGO oxidation processes. Next, an rGO–Ag-based nanocomposite was prepared using a simple mixing process, and the prepared rGO–Ag nanocomposite was tested against *S. aureus* under blue light irradiation. Their data suggested that the prepared rGO–Ag nanocomposite effectively inhibited *S. aureus* within 30 min of blue light exposure [76]. Ding et al. synthesized CuS–g-C_3_N_4_–based composite materials using a simple hydrothermal process and tested them against both *E. coli* and *S. aureus* bacterial strains. Their data indicated that the synthesized CuS–g-C_3_N_4_-based composite materials effectively killed/inhibited bacterial strains under solar irradiation. Moreover, increasing reaction temperatures and the amount of CuS increased the photo-antibacterial activity due to the increased recombination of photo-generated electron–hole pairs. Additionally, ROS generation increased the bacterial strains’ cellular disruption [77]. Tan et al. synthesized Ag–rGO-based composite materials for controlling bacterial infection under NIR exposure. Their results showed that the prepared Ag–rGO-based composite effectively killed/inhibited *E. coli* and MDR *Klebsiella pneumonia* (*K. pneumonia*) within 5 min of NIR exposure. The NIR exposure induced cellular membrane disruption via ROS generation [78]. 

Qian et al. Zn doped g-C_3_N_4_ incorporated with Bi_2_S_3_ (Zn–gC_3_N_4_–Bi_2_S_3_) using polymerization and electrostatic absorption. Their results demonstrated that the Zn–g-C_3_N_4_–Bi_2_S_3_ effectively controlled the *P. aeruginosa* and *S. aureus* infection within 15 min of exposure. The unusual photo-responsive antibacterial activity of the Zn–g-C_3_N_4_–Bi_2_S_3_ composite material was mainly due to the lower recombination rate and higher photo-generated current; the authors concluded that the material might effectively be used for tracheal repair [79]. Table 2 summarizes different metal-doped 2D-NMs and their photo-antibacterial activity. The data mentioned above and Table 2 indicate that the integration of metal within the 2D-NMs might enhance the photo-antibacterial activity mainly by significantly enhancing the electronic properties, porosity, and photon adsorption ability, as well as narrowing the band gap values, resulting in a lower recombination rate and higher photo-generated current and thus, enhanced photo-antibacterial activity. Moreover, the synthesis process also affects the electronic structures of metal–2D-NMs, suggesting that similar materials subjected to different synthesis processes might change their photo-antibacterial activity. 

### 4.2. Photo-Antibacterial Activity of 2D-NM-Based Hybrid Materials

At present, 2D-NMs and metal-doped 2D-NMs are of great interest to researchers due to their exceptional characteristics. Remarkably, incorporating metals within different 2D-NMs augments the materials’ applicability, including for photo-antibacterial applications. However, some of the most significant challenges involve the stability of 2D-NMs, high photo-antibacterial activity, and the leach-out ability of the metal ions from metal-doped 2D-NMs. Incorporating polymers, metals, carbon, and other 2D-NMs within the 2D-NMs (i.e., 2D-NM-based hybrid materials) might resolve such issues. Numerous 2D-NM-based hybrid materials have been used to control bacterial infections under solar irradiation. For instance, Xiao et al. synthesized poly-thiophenes that incorporated rGO (P–rGO) using the grafting method for controlling bacterial infections under solar irradiation. Their results show that the P–rGO inhibited/killed 100% *E. coli* within 10 min of irradiation due to significantly enhanced optical absorption (UV to NIR) and photo-generated electron transfer [80]. Using a condensation process, Qian et al. synthesized glycol-chitosan that incorporated carboxyl graphene (GCCG). The prepared GCCG effectively controlled the focal infection by killing *E. coli* and *S. aureus* within 10 min of NIR exposure. Moreover, effective contact and high surface area might enhance the photo-responsive ability [81]. Tan et al. synthesized BP nanosheets (BPS) using a simple liquid exfoliation process and encapsulated them with poly (4-pyridone methyl styrene) (PPMS) to produce a BPS–PPMS composite. Their results show that the prepared BPS–PPMS composite effectively inhibited/killed both *E. coli* and *S. aureus* bacterial strains within 10 min of exposure by generating ROS. Additionally, the prepared BPS–PPMS composite had high biocompatibility, and incorporating the PPMS polymer significantly improved the BPS chemical stability [82]. Figure 5 shows a schematic illustration of the Ti–PPMS–BS-based photo-antibacterial activity. Chen et al. synthesized peptide-conjugated dopamine–rGO (PD–rGO) using the oxidation and mixing process. Their results show that the prepared PD–rGO-based composite materials effectively killed *S. aureus* bacterial strains within 10 min of NIR exposure. Additionally, incorporating the protein within the PD–rGO-based composite materials significantly enhanced the therapeutic efficacy of the composite [83].

Budimir et al. synthesized a Kapton–Au–polyethylene-based composite with rGO (K–Au–PE–rGO) for controlling bacterial infection under NIR irradiation. Their results show that the prepared K–Au–PE–rGO composite material effectively killed/inhibited *E. coli* and *S. aureus* within 10 min and *Staphylococcus epidermidis* (*S. epidermidis*) within 30 min of NIR exposure. The electron-rich amino group improved photo-thermal efficiency [84]. Zhu et al. synthesized chitosan–MoS_2_–Ag that was deposited onto a Ti (CMAT) surface. Their results show that the Ag–MoS_2_ exhibited higher photo-antibacterial activity compared with the CMAT-based composite against both *E. coli* and *S. aureus* bacterial strains due to rapid electron transfer, resulting in high ROS production. Nonetheless, the prepared CMAT-based composite also effectively inhibited both *E. coli* and *S. aureus* with 98.6% and 99.7% inhibition, respectively [85]. Yuan et al. synthesized MoS_2_–PDA–RGD for controlling a bacterial infection under solar irradiation. Their results show that the prepared MoS_2_–PDA–RGD composite effectively controlled bacterial infection under NIR irradiation due to the generation of ROS and oxidative stress [86]. Figure 6 shows the SEM and photographic images of the antibacterial activity of MoS_2_–PDA–RGD. 

Zhang et al. synthesized CuS–MoS_2_-based hydrogel and tested it against wound-healing and controlling bacterial infections. Their data indicated that the CuS–MoS_2_-based hydrogel inhibited bacterial growth under solar light irradiation [87]. Xiang et al. synthesized ZnO–carbon dots–C_3_N_4_ (Z–C–C_3_N_4_)-based composite materials to inactivate bacterial strains and improve wound-healing ability. Their results indicate that the Z–C–C_3_N_4_-based composite materials effectively controlled bacterial (*E. coli and S. aureus*) infection with 99.97 and 99.99% inhibition, respectively, due to the generation of ROS and hyperthermia [88]. Zhang et al. synthesized a composite incorporating red phosphorus (RP) and GO (RP–GO) composite. Their results indicate that the prepared RP–GO-based composite effectively inactivated bacterial (*E. coli* and *S. aureus*) strains with 99.99% inhibition within 20 min of solar irradiation [89]. Li et al. synthesized Bi_2_S_3_/Ti_3_C_2_T*_x_*-based composite materials using a simple etching process for photo-antibacterial activity. Their results indicate that the prepared Bi_2_S_3_/Ti_3_C_2_T*_x_*-based composite materials effectively killed/inhibited *S. aureus* and *E. coli* with 99.86% and 99.92% inhibition at 10 min of irradiation, respectively, by increasing the production of ROS. Incorporating Ti_3_C_2_T_x_ in the Bi_2_S_3_ increased the impurity level, reducing the band gap value and increasing the photo-generated electrons, thereby generating high ROS [90] (Figure 7). Zhang et al. synthesized phosphorus (P)-doped MoS_2_–g-C_3_N_4_-based composite materials for the inactivation of bacteria under solar light irradiation. Their results indicate that the prepared P–MoS_2_–g-C_3_N_4_-based composite materials completely inhibited *E. coli*, whereas P–MoS_2_ showed 61.6% and g-C_3_N_4_ showed 44.7% inhibition. The prepared P–MoS_2_–g-C_3_N_4_-based composite materials improved the active site, thereby creating high photo-antibacterial activity [91]. Cao et al. synthesized chlorine-loaded chitosan-functionalized MoS_2_ (CC–MoS_2_) using simple exfoliation and mixing processes for photodynamic antibacterial activity. Their results indicate that the prepared CC–MoS_2_-based composite effectively killed/inhibited bacterial strains within 5 min of laser light exposure. Moreover, the data indicated that the CC–MoS_2_-based composite is highly efficient compared with other cationic composites [92]. 

Table 3 summarizes the 2D-NM-based hybrid materials and their photo-antibacterial application. Some salient observations can be made based on the literature above and Table 3. Indeed, metal incorporation within the 2D-NMs significantly enhanced the photocatalytic or photo-antibacterial activity, addressing a significant drawback of 2D-NMs alone. The main advantage of 2D-NM-based hybrid materials is the ability to tune the desired characteristics of the materials based on their applicability. These 2D-NM-based hybrid materials, as photo-antibacterial agents that are easily reusable up to 3-5 times without any significant difference in efficiency and that do not induce bacterial resistance, have major advantages over commercially available antibiotics. Moreover, due to their synergetic effects, 2D-NM-based hybrid materials show significantly high photo-antibacterial activity, exceptional biocompatibility, and are environmentally friendly. Additionally, the photo-antibacterial activity of 2D-NM-based hybrid materials mainly generates ROS, which leads to the disruption of the cellular membrane and subsequently the death of bacteria. 

## 5. Strategy to Improve Photo-Antibacterial Efficiency

The photo-antibacterial efficiency directly depends on the design of photocatalytic materials that can tune their band gap value and photo-absorption ability. Designing efficient photocatalytic materials is one of the most significant challenges of today. Researchers unceasingly focus on newer photocatalytic materials by changing their structures via incorporation or inserting dopants (metal/non-metals) that increase/decrease the band gap value, subsequently producing high photocatalytic activity [48,94]. Usually, the doping process introduces foreign elements (metal/non-metals) within semiconductor materials that significantly enhance a material’s applicability. There are mainly three types of dopants, (1) metals, (2) non-metals, and (3) co-dopants. These dopant materials directly tune the properties of photocatalyst materials in various ways, such as by (1) improving surface and interface characteristics, (2) tuning the band gap value and electronic configuration, (3) reducing the recombination rate of photo-generated electrons, or (4) improving the stability of the 2D-NMs. The perfect incorporation of dopants within the photoactive materials might promote the adsorptive ability to enhance photocatalytic efficiency via more internal active sites, uniform pore size, higher surface area, band gap value, or more polar environments. Incorporating metal or inorganic dopants might improve pore size distribution, resulting in higher photocatalytic activity. The porosity plays a crucial role in intra-particle diffusion for adsorption, and photo-degradation is one of the most important factors that directly affect the performance and define the synergy between factors [95,96,97,98,99].

Non-metal elements like nitrogen, carbon, boron, sulfur, fluorine, and chlorine can efficiently enhance visible light absorption performance, thereby narrowing the bandgap and the shift of the absorption edge, producing higher photocatalytic activity. Nitrogen is one of the most efficient candidates for visible light-induced antibacterial photo-response, as N provides a red shift of the absorption edge. Incorporating N by substituting O leads to a low bandgap value because the hybridization of N and O 2p levels results in a redshift, enhancing photo-absorption ability. Other dopants like S, B, C, and P demonstrate excellent photo-responsive skills by narrowing the band gap value, resulting in a high photo-antibacterial response [100,101,102]. It is essential to mention that enhanced photo-response mainly depends on the impurity level of non-metal dopants, which are usually above the valence band (VB) and lead to the extended optical absorption edge of photo-responsive materials.

Non-metal dopants tend to acquire isolated sites and reduce agglomeration within a semiconductor material, positioning it above the VB maximum and enhancing photo-response ability. Additionally, non-metal doping has several advantages that are similar to metal doping, and co-doping (metal–non-metal or metal–metal doping and non-metal doping) is another way to improve photo-antibacterial response by modulating the band gap up to a desired level. In other words, metal and non-metal doping are promising strategies to enhance the photo-induced antibacterial effect. The collaborative outcome of reducing the band gap is enhanced photo-absorption ability, less recombination of photo-generated holes and electrons, and a large population of charge carriers, subsequently enhancing the photo-antibacterial response [49,103,104,105,106,107,108]. 

Despite the numerous parameters involved, such as light source, reactor shape, reaction temperature, and distance between lamp and reactor, it is obvious that doping of carbon nitride with transitional metals, mainly Pt, significantly improves the photo-responsive ability compared with that of the non-metals like S, P, and N [109,110,111]. Another way to improve the photo-responsive ability of materials is self-doping, which narrows the band gap value of photocatalyst materials. The self-doping process is considered a promising method for tuning band gap values and electronics properties with a marginal morphological alteration. Moreover, self-doped photocatalysts provide enhanced structural characteristics with high surface area and inter-connectivity of the materials, thereby improving electron transport and subsequently producing high photo-responsive ability [112,113,114,115,116]. In general, the incorporated material (metal/non-metal/polymers) and self-doping possess a high extinction coefficient of light absorption with high carrier mobility, high porosity, surface area, and visual transparency to increase photo response, especially for bacterial degradation.

## 6. Biocompatibility of the 2D-NMs and Their Hybrid Materials

Researchers show ongoing interest in 2D-NMs for a variety of biomedical applications due to their exceptional physicochemical characteristics. The biomedical applications of 2D-NMs, such as drug delivery, nanomedicine, bio-imaging, and photo-thermal/photodynamic therapy, strongly motivate investigation of their biocompatibility. Researchers unceasingly focus on unraveling the interaction of 2D-NMs with cells/tissue/organs and their toxicity, which is essential before biological application. Therefore, researchers focus on synthesizing 2D-NMs in terms of their biocompatibility. The biocompatibility of 2D-NMs might be tuned by incorporating polymers, surface functional groups, and other materials. Studies have advised that 2D-NMs show insignificant toxicity, while others show some toxicity. It is important to mention that the design and synthesis process might affect the toxicity of 2D-NMs and their hybrid materials via external conditions (ionic strength and temperature) and surfactants used during the synthesis process. The toxicity of 2D-NMs depends on the types of materials, their size, shape, and surface charge, and the types of cells involved [117,118,119,120]. The interaction of 2D-NMs with biological environments such as cell lines, bacteria, animals, and biomolecules has been examined in the genetics and cellular toxicity contexts. The biocompatibility of 2D-NMs might depend on the type of 2D-NM and its surface properties and surface group. Therefore, it is necessary to understand the genetic/cellular toxicity of different 2D-NMs with different cells. Numerous 2D-NMs like graphene, GO, WS_2_, MXene, and MoS_2_ have been used to ascertain biocompatibility against different cells. For instance, Lehner et at. synthesized GO using a simple oxidation process, and the GO surface was functionalized with an amine group. Next, the GO–amine-group was coupled with DNA. The surface amine group significantly enhanced the biocompatibility of the GO (Figure 8) [121]. Another study synthesized Si_3_N_4_ nanosheets and tested them against the p53 tumor suppressor. The data indicated that the Si_3_N_4_ nanosheets have high biocompatibility [122]. Yang et al. synthesized graphene incorporated with polyethylene glycol (PEG–graphene) and ascertained the pharmacokinetics in vivo. The data suggested that the PEG–graphene distribution over different time ranges has high biocompatibility, indicating its effective use in biomedical applications [123]. Zhang et al. synthesized dextran-incorporated GO (D–GO) for in-vivo tracking of radioactive isotopes. The data indicate that the surface functionalization of GO using dextran significantly reduced toxicity compared with GO [124]. Xie et al. synthesized lipid-functionalized WS_2_ for photo-chemotherapy. The data indicate that lipid functionalization reduced toxicity and mainly depends on the concentration. Another group focused on WS2 doped with bovine serum albumin (BSA) (BSA–WS2) to improve theranostic ability. The data indicate that BSA–WS_2_ has high biocompatibility compared with WS_2_ [125]. PEG–WS_2_ was also synthesized for the photo-thermal treatment of tumors. The data indicate that incorporating PEG within the WS2 improves biocompatibility and effectiveness [126]. The study reports that the incorporation of surface functional groups, polymers, hydrophilic groups, biomolecules, and carbon within the 2D-NMs enhanced biocompatibility. Numerous studies reported the insignificant toxicity of 2D-NM-based hybrid materials against different cell lines, whereas higher cellular toxicity was observed at higher doses. Moreover, different materials show different biocompatibility levels and attributed cellular/genetic toxicity, depending on the materials and cell lines [80,127,128,129,130,131,132,133,134,135]. 

## 7. Artificial Intelligence (AI) in Photo-Antibacterial Activity

Researchers have dedicated remarkable effort to developing newer semiconductor materials, including 2D-NM-based hybrid materials that are proficiently used for controlling bacterial infection. These photo-antibacterial agents depend on semiconductor characteristics such as crystallinity, size, shape, surface area, and porous texture. Furthermore, dosage, pH, pollutant concentration, light intensity, and wavelength can also affect photocatalytic activity. However, enumerating the efficacy of photocatalysts for controlling bacterial infection is a significant challenge [136,137,138,139,140,141,142,143]. In this respect, designing experiments and optimizing semiconductor materials using AI might produce next-generation tools that can minimize the cost of experiments.

Machine learning (ML) is a branch of AI that could provide a data-driven process for the effectual investigation and extrapolation of photocatalyst performance. Usually, ML-based models in the literature are used to generate and subsequently manage experimental designs [139,144,145]. Numerous ML-based models have been developed to optimize photocatalyst materials for the photo-degradation of various contaminants, including bacteria. For instance, the literature contains examples of photocatalytic water splitting, solar photocatalysts, bismuth–ferrite for the removal of malachite green, octahedral 2D materials, antibacterial activity, nanoparticles for antibacterial activity, the degradation of perfluorooctanoic acid and water contaminants, the TiO_2_-based photo-degradation of water contaminants, the TiO_2_-based photo-degradation of phenol, Fe-based tri-composite for the photo-degradation of malachite green, research on the photocatalytic property of BiFeO_3_–WO_3_, 2D material components for photocatalysts, and photocatalytic membranes [45,46,146,147,148,149,150,151,152,153,154,155,156,157]. Using ML, we can save time, experimental costs, and manpower that would otherwise be spent in wet-lab experiments. ML is an evolving area that provides a newer solution for photo-antibacterial materials assessment. Artificial neural networks (ANN) are extensively used to envisage photocatalyst (carbon, metal, polymer, ceramic, and composite) material characteristics and assist in designing/discovering novel photocatalyst materials. However, a limited model with a similar structure and pollutant remains a concern. To the best of our knowledge, no studies have yet directly focused on photo-antibacterial activity, as most of the studies focus on designing photocatalysts and optimizing environmental pollutants/bacteria. Therefore, extensive research is required to develop a newer model with different material properties, structures, and pollutants. That said, these ML-based models demonstrate their usefulness in providing realistic solutions for synthesizing materials. Undoubtedly, the integration of ML in material science, chemistry, and biology will result in the enhanced applicability of these proposed materials with minimal cost, which might be beneficial for developing research and managing societal problems. 

## 8. Conclusions

In summary, this review article provides a platform for researchers/academicians to understand the development of photo-responsive 2D-NM-based hybrid materials, especially those with photo-antibacterial activity, to control bacterial infection in a smarter way. These photo-antibacterial agents effectively treat bacterial infection within a shorter exposure time without any adverse effects—most importantly, without causing bacterial resistance. Using such photo-antibacterial agents, we can eliminate bacterial infection on the surface of biological implants and infection sites and promote the wound-healing process. 2D-NM-based hybrid materials are emerging materials that efficiently kill/inhibit bacteria due to their exceptional properties, including photo-adsorption and lower band gap values. The lower band gap value improves the photon absorption ability, resulting in high photo-antibacterial activity. Metal doping within 2D-NMs might enhance the photo adsorption ability as well. Typically, incorporating metals within the 2D-NMs increased active sites, caused identical pore-size distribution, and narrowed the band gap values, thereby creating high photo-responsive ability. Incorporating metals, polymers, carbon, and surface functional groups within the 2D-NMs significantly enhanced photo adsorption and biocompatibility. Additionally, 2D-NM-based hybrid materials significantly exalted the excitation of electron–hole pairs, separation of charge, redox reaction on catalyst surfaces, and generated ROS and subsequently the cell death of bacterial strains. The biocompatibility of the 2D-NM-based hybrid materials against different cell lines showed insignificant toxicity, whereas higher cellular toxicity was observed at higher doses. The different materials showed a different level of biocompatibility, which was attributed to cellular/genetic toxicity that depended on the materials and cell lines. Furthermore, 2D-NM-based hybrid materials were reusable as photo-antibacterial agents for up to 3–5 times without any significant difference in efficiency and without inducing bacterial resistance, which is one of their major advantages over commercially available antibiotics. Therefore, we should consider various methods for improving the photo-antibacterial activity of next-generation 2D-NM-based hybrid materials in terms of designing metals, materials composition, and biocompatibility as much as possible. We also need to develop newer 2D-NM-based hybrid materials with enhanced photo-adsorption ability and biocompatibility that can efficiently filtrate water and control bacterial infection. ML in photocatalysts offers new insight by reducing the cost of experiments and manpower due to their optimization of photocatalyst dosage, pH, the concentration of pollutants, design of materials, and temperature. An ML-based model might provide newer pathways to developing a reactivity model of catalysts and their association with structure (size/shape), reaction conditions (temperature/pressure/agitation), the composition of materials (carbon/metal/polymers/ceramics/composite), and pollutants (chemical/biological). Using such ML models, we could easily calculate the exact amount of catalyst and its reactivity with pollutants at respective conditions. Finally, we hope this review will inspire young researchers to develop next-generation 2D-NM-based hybrid materials for controlling bacterial infections and take innovative steps toward smart water-treatment technology.

## Figures and Tables

**Figure 1 antibiotics-12-00398-f001:**
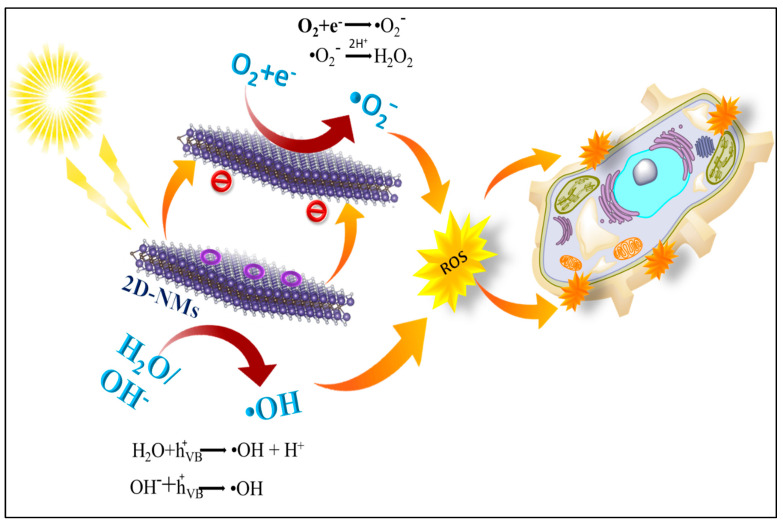
A schematic illustration of 2D-NM-based photo-antibacterial activity.

**Figure 2 antibiotics-12-00398-f002:**
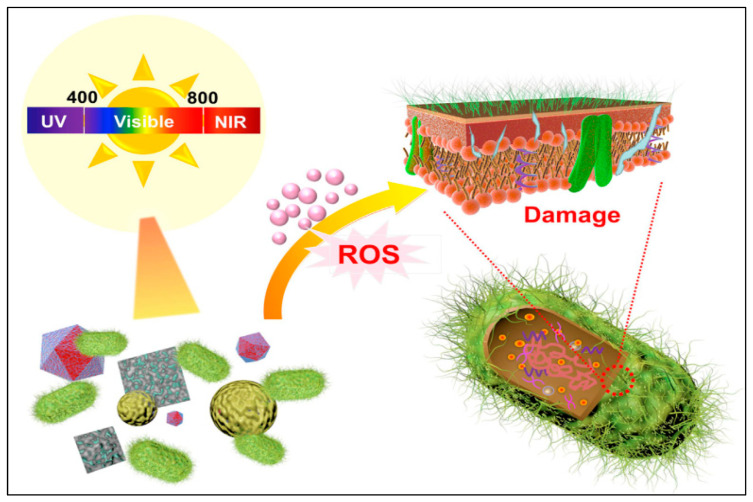
Photo-antibacterial activity of nanomaterials. The image was reproduced with permission [48].

**Figure 3 antibiotics-12-00398-f003:**
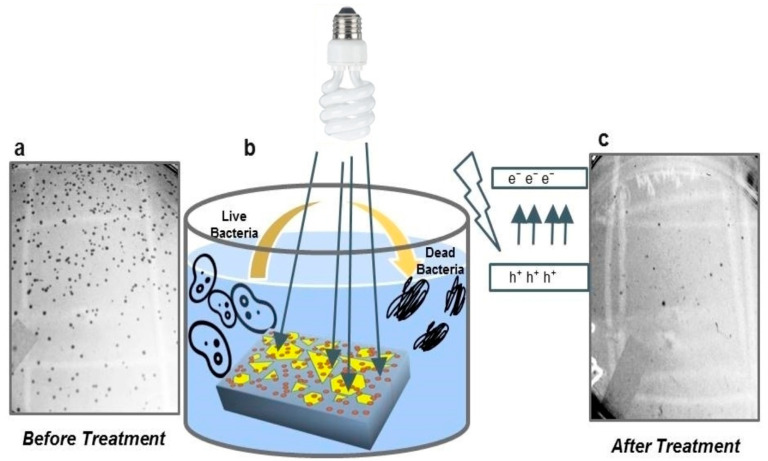
Schematic illustration of the photo-antibacterial activity of Ag–WS_2_ (**a**) before treatment, (**b**) Ag–WS_2_ exposed to the bacteria, and (**c**) after treatment. The image was taken with permission [68].

**Figure 4 antibiotics-12-00398-f004:**
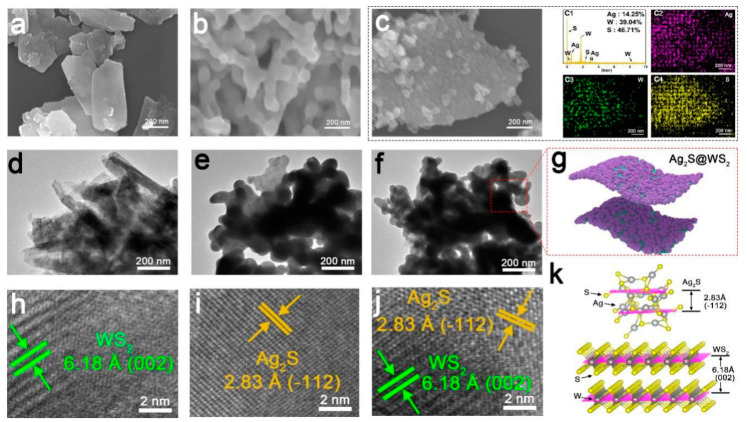
SEM, TEM, EDS, and crystal structure of Ag_2_S, WS_2_, and Ag_2_S–WS_2_. (**a**) WS_2_, (**b**) Ag_2_S, (**c**) Ag_2_S–WS_2_, (**d**) TEM image of WS_2_, (**e**) TEM images of Ag_2_S, (**f**) TEM image of Ag_2_S–WS_2_, (**g**) morphology of Ag_2_S–WS_2_, (**h**) HR-TEM image of WS_2_, (**i**) HR-TEM image of Ag_2_S, (**j**) HR-TEM image of Ag_2_S–WS_2_, and (**k**) crystal structure of Ag_2_S, and WS_2._ The image was reproduced with permission [73].

**Figure 5 antibiotics-12-00398-f005:**
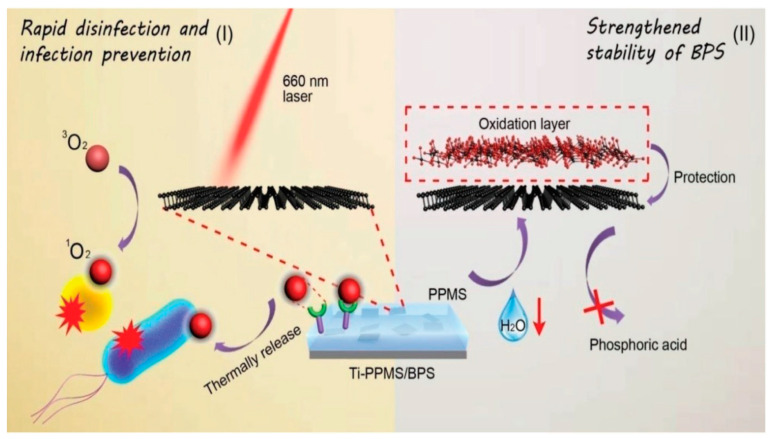
Schematic illustration of the Ti–PPMS–BS-based photo-antibacterial activity. The image was taken with permission [82].

**Figure 6 antibiotics-12-00398-f006:**
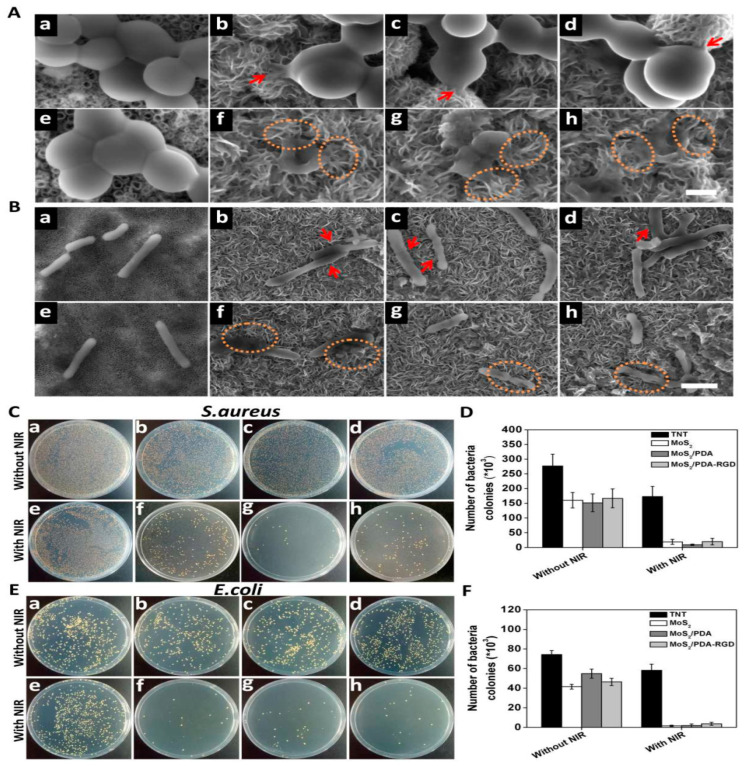
SEM and photographic images of the antibacterial activity of MoS_2_–PDA–RGD, (**A**) SEM images of *S. aureus* and (**B**) SEM images of *E. coli*, (**C**,**D**) photograph and number of colonies of *S. aureus* and (**E**,**F**) photograph and number of colonies of *E. coli*. (**a**) TNT, (**b**) MoS_2_, (**c**) MoS_2_-PDA, (**d**) MoS_2_-PDA-RGD, (**e**) TNT/NIR, (**f**) MoS_2_/NIR, (**g**) MoS_2_/PDA/NIR and (**h**) MoS_2_-PDA-RGD-NIR. The image was taken with permission [86].

**Figure 7 antibiotics-12-00398-f007:**
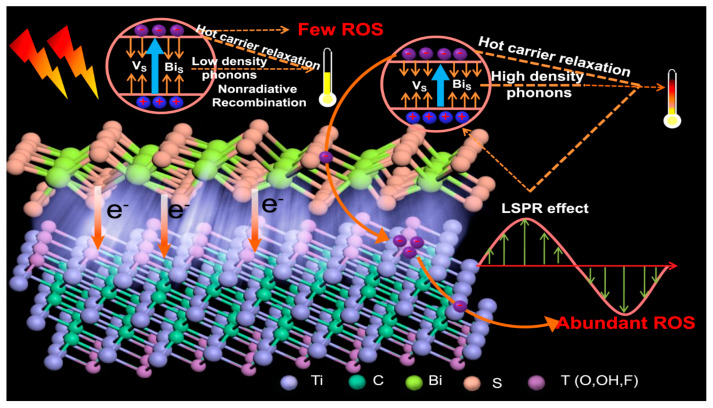
A schematic illustration of the photo-thermal mechanism of Bi_2_S_3_ and Ti_3_C_2_T*_x_*. The data was reproduced with permission [90].

**Figure 8 antibiotics-12-00398-f008:**
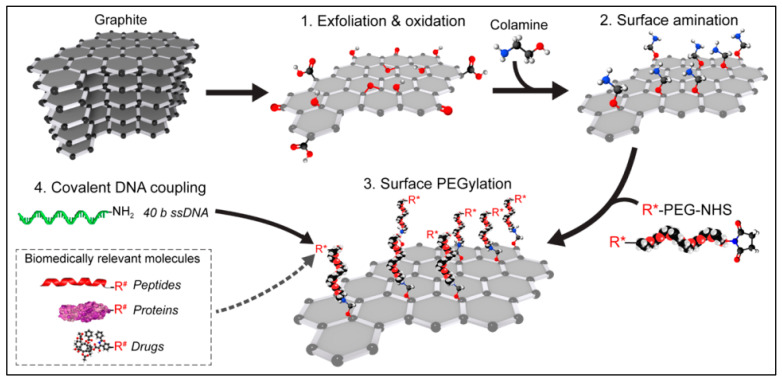
Schematic illustrating the synthesis of biocompatible GO. The image was reproduced with permission [121].

**Table 1 antibiotics-12-00398-t001:** Different 2D-NMs and their photo-antibacterial activity.

S. No.	2D-NMs	Synthesis Process	Bacteria	Inhibition Time (min)	References
1.	g-C_3_N_4_	Heating	*E. coli*	240	[60]
2.	BP	Exfoliation	*E. coli* and *S. aureus*	10	[61]
3.	MoS_2_	Li-ion intercalation	MDR *E. coli* and MRSA	15 and 25, respectively	[62]
4.	MoS_2_	Ultra-sonication, hydrothermal treatment, and intercalation	*E. coli*	180	[63]
5.	GO	Hummers method	19 types of bacteria	1440	[26]
6.	MoS_2_	Annealing	*E. coli*	10	[64]

**Table 2 antibiotics-12-00398-t002:** Metal-doped 2D-NMs and their photo-antibacterial activity.

S. No.	Metal–2D-NMs	Synthesis Process	Bacteria	Exposure Time (min)	References
1.	Ag–g-C_3_N_4_	Thermal polymerization	*E. coli*	90	[65]
2.	Ag–Si–rGO	Oxidation	*E. coli*	5	[72]
3.	Ag_2_S–WS_2_	In-situ	*E. coli* and *S. aureus*	20	[73]
4.	Ag–WS_2_	CVD	*E. coli*	180	[68]
5.	Ag–rGO	Oxidation and mixing	*E. coli* and *K. pneumonia*	10	[78]
6.	Ag–rGO	Green process	*S. aureus*	30	[76]
7.	AgWO_4_-gC_3_N_4_	Deposition–precipitation	*E. coli*	90	[66]
8.	GO–g-C_3_N_4_	Sonication	*E. coli*	120	[67]
9.	CuS–g-C_3_N_4_	Hydrothermal	*E. coli* and *S. aureus*	20	[77]
10.	GO–CuO	Thermal oxidation	*E. coli*	150	[70]
11.	V-QD–g-C_3_N_4_	Thermal	*Salmonella*	10	[71]
12.	MnO_2_–g-C_3_N_4_	-	*E. coli* and *S. aureus*	20	[74]
13.	CeO_2_–GO	Co-precipitation	*E. coli, S. aureus, S. typhi, P. aeruginos.*	-	[75]
14.	ZnO–GQDs	Heating	*E. coli* and *S. aureus*	5	[69]
15.	Zn–g-C_3_N_4_–Bi_2_S_3_	Polymerization and electrostatic absorption	*S. aureus* and *Pseudomonas*	15	[79]

**Table 3 antibiotics-12-00398-t003:** 2D-NM-based hybrid materials and their photo-antibacterial application.

S. No.	Hybrid Materials	Synthesis Process	Bacteria	Exposure Time (min)	References
1.	P–rGO	Reduction	*E. coli*	10	[80]
2.	GCCG	Condensation	*E. coli* and *S. aureus*	10	[81]
3.	BPS–PPMS	Exfoliation	*E. coli* and *S. aureus*	10	[82]
4.	MoS_2_–PD–RGD	Immobilization	*E. coli* and *S. aureus*	8	[86]
5.	GO–NH_2_	Oxidation	*E. coli* and *S. aureus*	10	[93]
6.	PD–rGO	Oxidation	*S. aureus*	10	[83]
7.	K–Au–PE–rGO	Exfoliation and reduction	*E. coli, S. aureus,* and *S. epidermidis*	30	[84]
8.	CMAT	-	*E. coli* and *S. aureus*	20	[85]
9.	CuS–MoS_2_-based hydrogel	Hydrothermal and freeze-thawing	*E. coli* and *S. aureus*	15	[87]
10.	Z–C–C_3_N_4_	Thermal polymerization	*E. coli* and *S. aureus*	15	[88]
11.	RP–GO	Hydrothermal	*E. coli* and *S. aureus*	20	[89]
12.	Bi_2_S_3_/Ti_3_C_2_T*_x_*	Etching	*S. aureus* and *E. coli*	10	[90]
13.	P–MoS_2_–g-C_3_N_4_	Layer-by-layer	*E. coli*	-	[91]
14.	CC–MoS_2_	Sonication and mixing	*E. coli*	5	[92]

## Data Availability

Not applicable.

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
