# Peer review of "Photo-Antibacterial Activity of Two-Dimensional (2D)-Based Hybrid Materials: Effective Treatment Strategy for Controlling Bacterial Infection"

_antibiotics, 2023, doi:10.3390/antibiotics12020398_

Round 1
Reviewer 1 Report
Dear auhtors.
Find attached the comments related to the review

Author Response
Reviewer:1
Comments and Suggestions for Authors
The authors wrote a review regarding the use of two-dimensional nanomaterials for bacterial disinfection. They analyze and present the studies made in this field underlining all the progresses in terms of photo antibacterial activity thanks to the development of new materials and synthesis strategies. The review is well articulated, and all the sections well discussed. The authors show multiple examples of the application of different materials for antibacterial activity. The sections are properly divided, tables and images showed are clear. I would suggest publishing this paper after minor revisions: My suggestion is to add a small generic paragraph that shows the advances in the synthesis and applications of hybrid materials before analyzing the antibacterial activity: - Molecules 2022, 27, 6828. https://doi.org/10.3390/molecules27206828
Reply: Authors thank the reviewer for appreciating our study and suggesting that the quality of the manuscript might be improved. As per the reviewer’s suggestion, we have now incorporated advances in synthesizing and applying hybrid materials in the revised manuscript.
Reviewer 2 Report
This manuscript focuses on the photo-antibacterial activity of two-dimensional (2D) based hyrid materials.
This is an interesting work; Nevertheless some minor revisions are needed in order to publish this work.
1. Could the authors add more information regarding the bacteria studied in the literature? Are there any more bacteria studied?
2. Is there any work studying the re-usability of their samples?
3. A few more references could be added
4. Some minor syntax errors could be corrected
Author Response
Reviewer:2
Comments and Suggestions for Authors
This manuscript focuses on the photo-antibacterial activity of two-dimensional (2D) based hyrid materials. This is an interesting work; Nevertheless some minor revisions are needed in order to publish this work.
Reply: Authors, thank the reviewer for appreciating our study. I hope the revised version might be acceptable for publication.
Q1. Could the authors add more information regarding the bacteria studied in the literature? Are there any more bacteria studied?
Reply: Authors, thanks to the reviewer’s comments and suggestions. We have re-check the literature and observed that most of the studies, especially photo-antibacterial studies focus on E. Coli and S. aureus, and very few studies focus on S. typhi, P. aeruginosa, K. pneumonia, Pseudomonas, Salmonella, MDR E. Coli & MRSA. We almost cover all bacterial strains in this study. In this aspect, there are no changes in the manuscript.
Q2. Is there any work studying the reusability of their samples?
Reply: Authors thanks to reviewer’s comments. Yes. We already mentioned the reusability of some materials up to 3-4 times in the text. For more clarity, we incorporate the reusability of the materials in the revised manuscript.
Q3. A few more references could be added.
Reply: We have now incorporated more references in the revised manuscript.
Q4. Some minor syntax errors could be corrected.
Reply: Corrected.
Reviewer 3 Report
In this review paper, Talreja et al summarized the recent progress on the photo-antibacterial study of 2D based hybrid materials. This review article provides a platform for researchers/academicians to understand the development of photo-responsive 2D-NMs-based hybrid materials, especially for photo-antibacterial activity, to control bacterial infection in a more effective way. I recommend its acceptance in Antibiotics after addressing the following comments.
1) The title can be more concise and precise, which might can be revised as “Photo-antibacterial Activity of Two-dimensional (2D)-based Hybrid Materials: A Review.”
2) “2D nanomaterials” is suggested added as the key words.
3) Although 2D materials might have photo-antibacterial activity, however, the discussions on the toxicity of 2D materials themselves are missing. I would like to suggest the authors to provide some discussions and related references on the issues of toxicity of 2D materials.
4) The ordinal numeration of subtitles of part III “ 3. Photo-antibacterial activity” are wrong, e.g., “2.1.2. D-NMs”, “2.2. Metal doped 2D-NMs”, which should be carefully checked.
5) The outlook and perspective should be strengthened in Conclusion part.
Author Response
Reviewer:3
Comments and Suggestions for Authors
In this review paper, Talreja et al summarized the recent progress on the photo-antibacterial study of 2D based hybrid materials. This review article provides a platform for researchers/academicians to understand the development of photo-responsive 2D-NMs-based hybrid materials, especially for photo-antibacterial activity, to control bacterial infection in a more effective way. I recommend its acceptance in Antibiotics after addressing the following comments.
Reply Authors, thanks to the reviewers for appreciating our study.
Q1. The title can be more concise and precise, which might can be revised as “Photo-antibacterial Activity of Two-dimensional (2D)-based Hybrid Materials: A Review.”
Reply: Thanks to the reviewer’s comments and suggestions. We respect the reviewer’s view, but we believe that photo-antibacterial activity might become an effective treatment strategy for controlling bacterial infection in the future; in this regard, we retain this title in the revised manuscript.
Q2. “2D nanomaterials” is suggested added as the key words.
Reply: Incorporated.
Q3. Although 2D materials might have photo-antibacterial activity, however, the discussions on the toxicity of 2D materials themselves are missing. I would like to suggest the authors to provide some discussions and related references on the issues of toxicity of 2D materials.
Reply: Thanks to the reviewer’s comments and suggestions. The author's presentation might confuse the reviewers, as we already discuss the toxicity aspect in section-5 (Biocompatibility of 2D-NMs). For more clarification, we revised this section and incorporated new references in the revised manuscript.
Q4. The ordinal numeration of subtitles of part III “ 3. Photo-antibacterial activity” are wrong, e.g., “2.1.2. D-NMs”, “2.2. Metal doped 2D-NMs”, which should be carefully checked.
Reply: Corrected.
Q5. The outlook and perspective should be strengthened in Conclusion part.
Reply: Thanks to the reviewer’s comments and suggestions. We have now revised the conclusion part in this aspect.
Reviewer 4 Report
In this work, Talreja et al present a review about the photo-antibacterial activity of two-dimensional (2D) nanomaterials (2D-NMs), including principle of photo-antibacterial activity, synthesis of novel 2D-NMs, biocompatibility of 2D-NMs and application of machine learning in this area. However, several major questions have not been addressed and some in-depth discussions are missing. Further, an improved organization and extensive language edits are needed to make this paper clearer.
1 I suggest authors put ‘3. Photo-antibacterial activity’ and ‘4. Role of oxygen in photo-antibacterial activity’ before ‘2. Strategy to improve photo-antibacterial efficiency’. As for the reader, it will make more sense to understand what is photo-antibacterial activity and its principle before moving to know how 2D-NMs are used in this area.
2 The content from line 169 to line 404 is more related to ‘2. Strategy to improve photo-antibacterial efficiency’ rather than ‘3. Photo-antibacterial activity’, I suggest authors change the location of this content.
3 I suggest authors merge ‘3. Photo-antibacterial activity’ and ‘4. Role of oxygen in photo-antibacterial activity’. As in 3, authors have already mentioned ROS but didn’t give further explanations, which make this part unclear and incomplete.
4 I suggest authors reorganize ‘5. Biocompatibility of the 2D-NMs’, as currently there are too many contents listed separately without clear connections. I suggest authors reduce the discussion about the importance of biocompatibility, as this is an obvious requirement for any biology application related materials, but discuss more about the biocompatibility of 2D-NMs and strategies of improving biocompatibility of 2D-NMs. To make a profound discussion, more examples are needed, and more importantly all these examples need to be well connected rather than listed separately.
5 I suggest authors add some examples to show the efficiency advantage of AI-based methods to ‘Artificial intelligence (AI) in photo-antibacterial activity’. It is especially meaningful if these examples show time consumption comparison between AI methods and traditional methods.
6 Though 2D-NMs are the most important term in this review, authors haven’t given any introduction to this concept, like what 2D-NMs are and how they are different from other materials.
7 As authors suggest, one major advantage of photo-antibacterial activity is the low chance of developing bacterial resistance. Authors mentioned the possible reason for this low chance is the rapid efficacy. However, antibiotics also kill bacteria rapidly but bacteria still develop resistance to them. Further, in reality bacteria usually live in the form of bacterial biofilm that they can construct complex structures and cooperate with each other, which also allow them to develop different resistance strategies. Authors need to provide an in-depth discussion to clear these concerns and to better support their opinion.
8 line 119 ‘a low bandgap value because the hybridization of N and O 2p levels results in a redshift, enhancing photo-absorption ability. ‘Authors haven’t explained why a low bandgap value will lead to better photo-absorption ability. As Authors mentioned it many times, I suggest authors give a detailed explanation about it.
9 line 130-131 ‘In other words, metal and non-metal doping is a promising strategy to enhance the photo-induced antibacterial effect.’ Only non-metal doping examples are shown before this conclusion. To support this conclusion, examples for metal doping are also needed.
10 line 134-137 ‘The incorporated …’, this part is not well connected with previous content, authors need reorganized this part and add more discussion, examples and references for this part.
11 line 203-206 ’The data mentioned above and in table 1…’, the data above line 203 or in table 1 may support that 2D-NMs are effective for photo-antibacterial activity but why it can be attributed to 2D-NMs‘ band gap values, high surface area, and exception electronics characteristics is not clear. Further explanations are needed.
12 line 418-419, this part is very confusing. I am not sure if this ‘another pathway’ is a way that is different from the previously mentioned two pathways or if it is a further extension of one of these two ways.
13 line 421-422 ‘In general, …, which might be possible in the absence of oxygen’, however, in the following lines, authors haven’t shown any examples of photo-antibacterial activity in the absence of oxygen, authors need to add these examples.
14 line 451 ‘Studies advised that the 2D-NMs show insignificant toxicity while others show some toxicity.’ I don’t understand it, especially I am not sure what ‘while others show some toxicity’ mean.
15 line 501 ‘as most of the studies focus on designing photocatalysts/environmental pollutants/bacteria to the best of our knowledge.’ I am not sure what ‘designing photocatalysts/environmental pollutants/bacteria’ mean, I may understand ‘designing photocatalysts’ but what ‘designing environmental pollutants/bacteria’ mean.
16 There are also many small mistakes:
Line 169 I believe it should be ‘2D-NMs’ not ‘2. D-NMs’ further why the numbering is ‘2.1’, as a content for ‘3. Photo-antibacterial activity’, isn’t it ‘3.1’, same question for line 211.
Line 205, ‘band gap’ not ‘bang gap’
Line 310 same issue ’2. D-NMs’
Line 345 ‘Staphylococcus epidermidis (S. epidermidis)’ , why the font size and style is different from other bacterial names.
There may be more of these kinds of mistakes, so I suggest the author carefully check the manuscript.
Author Response
Reviewer-4
Comments and Suggestions for Authors
In this work, Talreja et al present a review about the photo-antibacterial activity of two-dimensional (2D) nanomaterials (2D-NMs), including principle of photo-antibacterial activity, synthesis of novel 2D-NMs, biocompatibility of 2D-NMs and application of machine learning in this area. However, several major questions have not been addressed and some in-depth discussions are missing. Further, an improved organization and extensive language edits are needed to make this paper clearer.
Q1. I suggest authors put ‘3. Photo-antibacterial activity’ and ‘4. Role of oxygen in photo-antibacterial activity’ before ‘2. Strategy to improve photo-antibacterial efficiency’. As for the reader, it will make more sense to understand what is photo-antibacterial activity and its principle before moving to know how 2D-NMs are used in this area.
Reply: Authors, thanks to the reviewer’s comments and suggestions. Agreed. We have now revised this aspect.
Q2. The content from line 169 to line 404 is more related to ‘2. Strategy to improve photo-antibacterial efficiency’ rather than ‘3. Photo-antibacterial activity’, I suggest authors change the location of this content.
Reply: Authors, thanks to the reviewer’s comments and suggestions. I think that they are related to each other. In this section, we describe the different 2D-NMs and metal-doped-2D-NMs and its photo-antibacterial activity; in this aspect, there are no changes in the revised manuscript.
Q3. I suggest authors merge ‘3. Photo-antibacterial activity’ and ‘4. Role of oxygen in photo-antibacterial activity’. As in 3, authors have already mentioned ROS but didn’t give further explanations, which make this part unclear and incomplete.
Reply: Authors, thanks to the reviewer’s comments and suggestions. The separate section on oxygen's role in photo-antibacterial activity describes the importance of oxygen in the photocatalysis process. In our view, we retain this section as it is in the revised manuscript.
Q4. I suggest authors reorganize ‘5. Biocompatibility of the 2D-NMs’, as currently there are too many contents listed separately without clear connections. I suggest authors reduce the discussion about the importance of biocompatibility, as this is an obvious requirement for any biology application related materials, but discuss more about the biocompatibility of 2D-NMs and strategies of improving biocompatibility of 2D-NMs. To make a profound discussion, more examples are needed, and more importantly all these examples need to be well connected rather than listed separately.
Reply: Authors, thanks to the reviewer’s comments and suggestions. As per the reviewer’s suggestion, we have revised this section with more examples in the revised manuscript.
Q5 I suggest authors add some examples to show the efficiency advantage of AI-based methods to ‘Artificial intelligence (AI) in photo-antibacterial activity’. It is especially meaningful if these examples show time consumption comparison between AI methods and traditional methods.
Reply: Authors, thanks to the reviewer’s comments and suggestions. Usually, AI help to optimize material's dose, temperature, concentration, and pH. With the help of AI, we can easily consume time as well as experimental costs. We have now incorporated more references in this aspect in the revised manuscript.
Q6.Though 2D-NMs are the most important term in this review, authors haven’t given any introduction to this concept, like what 2D-NMs are and how they are different from other materials.
Reply: Authors, thanks for the comments. We have now incorporated a brief discussion about 2D-NMs in the introduction section.
Q7. As authors suggest, one major advantage of photo-antibacterial activity is the low chance of developing bacterial resistance. Authors mentioned the possible reason for this low chance is the rapid efficacy. However, antibiotics also kill bacteria rapidly but bacteria still develop resistance to them. Further, in reality bacteria usually live in the form of bacterial biofilm that they can construct complex structures and cooperate with each other, which also allow them to develop different resistance strategies. Authors need to provide an in-depth discussion to clear these concerns and to better support their opinion.
Reply: Authors, thanks for the comments. Usually, the photo-antibacterial activity offers newer possibilities for controlling bacterial infections from water using photocatalytic materials, cellular oxygen, and visible light irradiation. We have now revised this section in the revised manuscript.
Q8. line 119 ‘a low bandgap value because the hybridization of N and O 2p levels results in a redshift, enhancing photo-absorption ability. ‘Authors haven’t explained why a low bandgap value will lead to better photo-absorption ability. As Authors mentioned it many times, I suggest authors give a detailed explanation about it.
Reply: Authors, thanks for the comments. The lower band gap values significantly improve the absorption of a photon, thereby high photo-antibacterial activity. We have now incorporated this sentence in the revised manuscript.
Q9. line 130-131 ‘In other words, metal and non-metal doping is a promising strategy to enhance the photo-induced antibacterial effect.’ Only non-metal doping examples are shown before this conclusion. To support this conclusion, examples for metal doping are also needed.
Reply: Authors, thanks for the comments. Agreed. Our focus is metal-doped 2D-NMs, not non-metal-doped 2D-NMs. We have incorporated new references in the revised manuscript to support the above sentence.
Q10 line 134-137 ‘The incorporated …’, this part is not well connected with previous content, authors need reorganized this part and add more discussion, examples and references for this part.
Reply: Corrected.
Q11. line 203-206 ’The data mentioned above and in table 1…’, the data above line 203 or in table 1 may support that 2D-NMs are effective for photo-antibacterial activity but why it can be attributed to 2D-NMs‘ band gap values, high surface area, and exception electronics characteristics is not clear. Further explanations are needed.
Reply: Authors, thanks to the reviewer’s comments. We have now clarified the above statements in the revised manuscript.
Q12. line 418-419, this part is very confusing. I am not sure if this ‘another pathway’ is a way that is different from the previously mentioned two pathways or if it is a further extension of one of these two ways.
Reply: Authors, thanks to the reviewer’s comments. Yes, another pathway is a way that is different from previous pathways. For more clarification, we have revised this statement in the revised manuscript.
Q13. line 421-422 ‘In general, …, which might be possible in the absence of oxygen’, however, in the following lines, authors haven’t shown any examples of photo-antibacterial activity in the absence of oxygen, authors need to add these examples.
Reply: Authors, thanks to the reviewer’s comments. Indeed, a special case like Psoralens is sensitive to light. We have revised these sentences in the revised manuscript.
Q14. line 451 ‘Studies advised that the 2D-NMs show insignificant toxicity while others show some toxicity.’ I don’t understand it, especially I am not sure what ‘while others show some toxicity’ mean.
Reply: Authors, thanks to the reviewer’s comments. It seems that the author's presentation might create some confusion. For more clarification, we have revised the biocompatibility of the 2D-NMs section in the revised manuscript.
Q15. line 501 ‘as most of the studies focus on designing photocatalysts/environmental pollutants/bacteria to the best of our knowledge.’ I am not sure what ‘designing photocatalysts/environmental pollutants/bacteria’ mean, I may understand ‘designing photocatalysts’ but what ‘designing environmental pollutants/bacteria’ mean.
Reply: We have now corrected the revised manuscript.
Q16. There are also many small mistakes:
Line 169 I believe it should be ‘2D-NMs’ not ‘2. D-NMs’ further why the numbering is ‘2.1’, as a content for ‘3. Photo-antibacterial activity’, isn’t it ‘3.1’, same question for line 211.
Line 205, ‘band gap’ not ‘bang gap’
Line 310 same issue ’2. D-NMs’
Line 345 ‘Staphylococcus epidermidis (S. epidermidis)’ , why the font size and style is different from other bacterial names.
There may be more of these kinds of mistakes, so I suggest the author carefully check the manuscript.
Reply: We have now checked the entire manuscript and corrected linguistic errors.
Reviewer 5 Report
The manuscript entitled ”Photo-antibacterial activity of two-dimensional (2D) based hybrid materials: Effective treatment strategy for controlling bacterial infection” has in attention the possibilities to reduce bacterial infections by using two-dimensional materials.
The authors are kindly requested to consider the following commendable recommendations:
- carefully check the entire manuscript for the English language;
- consider including a description of the most used materials for different bacterial infections (table or figure) and give details in the text;
- re-organize the sections and/or firstly present the materials and thereafter present strategies for their improvement;
- check the numbering of the sections and sub-sections;
- check all abbreviations and insert their names where they are first appearing;
- table 1, if the inhibition time is in minutes, then do not include hours;
- table 2, re-organize depending on the metal in the materials;
- consider presenting the limitations of using these materials for water disinfection.
Author Response
Reviewer-5
Comments and Suggestions for Authors
The manuscript entitled ”Photo-antibacterial activity of two-dimensional (2D) based hybrid materials: Effective treatment strategy for controlling bacterial infection” has in attention the possibilities to reduce bacterial infections by using two-dimensional materials.
The authors are kindly requested to consider the following commendable recommendations:
Q1- carefully check the entire manuscript for the English language;
Reply: We have now checked the entire manuscript and corrected linguistic errors.
Q2- consider including a description of the most used materials for different bacterial infections (table or figure) and give details in the text;
Reply: Authors, thanks for the comment. We have now checked the entire manuscript and correct it.
Q3- re-organize the sections and/or firstly present the materials and thereafter present strategies for their improvement;
Reply: Corrected.
Q4- check the numbering of the sections and sub-sections;
Reply: Corrected.
Q5- check all abbreviations and insert their names where they are first appearing;
Reply: Corrected.
Q6- table 1, if the inhibition time is in minutes, then do not include hours;
Reply: Corrected.
Q7- table 2, re-organize depending on the metal in the materials;
Reply: Corrected.
Q8- consider presenting the limitations of using these materials for water disinfection.
Reply: We have now included the limitation of the materials.
Round 2
Reviewer 3 Report
I think the issues raised by the reviewers have been addressed.
Author Response
Reviewer:3
Comments and Suggestions for Authors
I think the issues raised by the reviewers have been addressed.
Reply: Authors, thanks to the reviewer for accepting our manuscript for publication.
Reviewer 4 Report
In the revised version, authors answered most of my question properly. But there are still some points need to be further addressed.
1 Now authors give more introduction about the characteristics of 2D-NMs, but it is still lacking a sentence to clearly define what is 2D-NMs. A sentence like '2D-NMs are a class of nanomaterial...' will be helpful.
2 Line 102-128 is a general introduction to Photo-antibacterial activity but Line 176 – 371 are all about application of 2D-NMs and their modified version in Photo-antibacterial activity. I will suggest you only keep line 102-128 for ‘Photo-antibacterial activity’ and put section 3 ‘Role of oxygen in photo-antibacterial activity’ after this. By that readers will get a better understanding of Photo-antibacterial activity before moving to the specific 2D-NMs. I will further suggest separate line 176-371 to a new section named something like ‘Photo-antibacterial activity of 2D-NMs’. You can start this new section by introducing why 2D-NMs are good materials for Photo-antibacterial activity and then go to your current line 176-371.
3 ‘2.2. Metal doped 2D-NMs’ and ‘4. Strategy to improve photo-antibacterial efficiency’ are a bit duplicate, as they both discuss strategy based on incorporation dopants. I suggest you modify section 4, maybe reduce content of incorporation dopants and add some discussion for other strategies that can improve photo-antibacterial efficiency.
4 For my previous Q7, authors indicate a few possibilities of how photo-antibacterial activity can control bacterial infections, but my question is why bacteria will not (or only has low possibility to) develop resistant to photo-antibacterial activity? Authors need addressed my concern.
5 line 121-122 is very confused, words are missing before ‘against’, authors need further check the manuscript for the language.
Author Response
Reviewer:4
Comments and Suggestions for Authors
In the revised version, authors answered most of my question properly. But there are still some points need to be further addressed.
Q1. Now authors give more introduction about the characteristics of 2D-NMs, but it is still lacking a sentence to clearly define what is 2D-NMs. A sentence like '2D-NMs are a class of nanomaterial...' will be helpful.
Reply: Authors, thanks for the comments and suggestions that might be improved the quality of the manuscript. As per the reviewer’s suggestion, we have now incorporated these sentences in the revised manuscript.
Q2. Line 102-128 is a general introduction to Photo-antibacterial activity but Line 176 – 371 are all about application of 2D-NMs and their modified version in Photo-antibacterial activity. I will suggest you only keep line 102-128 for ‘Photo-antibacterial activity’ and put section 3 ‘Role of oxygen in photo-antibacterial activity’ after this. By that readers will get a better understanding of Photo-antibacterial activity before moving to the specific 2D-NMs. I will further suggest separate line 176-371 to a new section named something like ‘Photo-antibacterial activity of 2D-NMs’. You can start this new section by introducing why 2D-NMs are good materials for Photo-antibacterial activity and then go to your current line 176-371.
Reply: Authors, thanks for the comments and suggestions. We have now changed accordingly in the revised manuscript.
Q3. ‘2.2. Metal doped 2D-NMs’ and ‘4. Strategy to improve photo-antibacterial efficiency’ are a bit duplicate, as they both discuss strategy based on incorporation dopants. I suggest you modify section 4, maybe reduce content of incorporation dopants and add some discussion for other strategies that can improve photo-antibacterial efficiency.
Reply: Authors, thanks for the comments and suggestions. We have now changed it accordingly in the revised manuscript.
Q4. For my previous Q7, authors indicate a few possibilities of how photo-antibacterial activity can control bacterial infections, but my question is why bacteria will not (or only has low possibility to) develop resistant to photo-antibacterial activity? Authors need addressed my concern.
Reply: Authors, thanks for the comments. Indeed, it is a very good question. The authors try to explain this question in the earlier revision but seem to fail. For more clarification, here is our explanation. Usually, photo-antibacterial activity involves the use of semiconductor materials to kill/inhibit the bacteria, whereas antibiotic molecules might lead to bacterial resistance by encouraging the survival of resistant bacterial strains. The photo-antibacterial agents kill bacteria by damaging the cellular membranes/other structures, thereby it tougher for the development of bacterial resistance. Moreover, photo-antibacterial agents often have a broader spectrum of antibacterial activity compared with antibiotic molecules, thereby difficult to develop bacterial resistance. It is important to mention here that the development of bacterial resistance is a complex process and occurs with prolonged or excessive use of any antibacterial agents. We have now incorporated the above discussion in the revised manuscript.
Q5. line 121-122 is very confused, words are missing before ‘against’, authors need further check the manuscript for the language.
Reply: Corrected.
Reviewer 5 Report
The authors have adequately addressed the comments of the reviewer.
Author Response
Reviewer: 5
Comments and Suggestions for Authors
The authors have adequately addressed the comments of the reviewer.
Reply: Authors, thanks to the reviewers for accepting our manuscript for publication.
Round 3
Reviewer 4 Report
The majority of my concerns were effectively addressed in the revised version, however, there is one remaining point that requires further attention.
The content added to line 469-472 is insufficient. Important parameters such as reactor shape, light source, and distance between lamp and reactor have not been previously or recently discussed, making it difficult to integrate this content with other sections. To bridge this knowledge gap, the authors should provide additional details and expand on the new content.
Author Response
Manuscript ID: antibiotics-2213584
Type of manuscript: Review
Title: Photo-antibacterial activity of two-dimensional (2D) based hybrid materials: Effective treatment strategy for controlling bacterial infection
Reviewer:4
Comments and Suggestions for Authors
The majority of my concerns were effectively addressed in the revised version, however, there is one remaining point that requires further attention.
Q1. The content added to line 469-472 is insufficient. Important parameters such as reactor shape, light source, and distance between lamp and reactor have not been previously or recently discussed, making it difficult to integrate this content with other sections. To bridge this knowledge gap, the authors should provide additional details and expand on the new content.
Reply: Authors, thank you for the comment and suggestions. As per the reviewer suggestions, we have now revised this section in the revised manuscript.